# A Precise and Robust Segmentation-Based Lidar Localization System for Automated Urban Driving

**Hang Liu [1], Qin Ye [1,*], Hairui Wang [2], Liang Chen [3] and Jian Yang [3]**

[1]  College of Surveying and Geo-Informatics, Tongji University, Shanghai 200092, China; liuhang@tongji.edu.cn (H.L.)
[2]  Beijing Momenta Technology Company Limited, Beijing 100190, China; hairui@momenta.ai
[3]  School of Computer Science and Engineering, Nanjing University of Science and Technology, Nanjing 210094, China; liangchen@njust.edu.cn (L.C.); csjyang@njust.edu.cn (J.Y.)
*  Correspondence: yeqin@tongji.edu.cn; Tel.: +86-1391-786-6679

**Abstract:** Real-time and high-precision localization information is vital for many modules of unmanned vehicles. At present, a high-cost RTK (Real Time Kinematic) and IMU (Integrated Measurement Unit) integrated navigation system is often used, but its accuracy cannot meet the requirements and even fails in many scenes. In order to reduce the costs and improve the localization accuracy and stability, we propose a precise and robust segmentation-based Lidar (Light Detection and Ranging) localization system aided with MEMS (Micro-Electro-Mechanical System) IMU and designed for high level autonomous driving. Firstly, we extracted features from the online frame using a series of proposed efficient low-level semantic segmentation-based multiple types feature extraction algorithms, including ground, road-curb, edge, and surface. Next, we matched the adjacent frames in Lidar odometry module and matched the current frame with the dynamically loaded pre-build feature point cloud map in Lidar localization module based on the extracted features to precisely estimate the 6DoF (Degree of Freedom) pose, through the proposed priori information considered category matching algorithm and multi-group-step L-M (Levenberg-Marquardt) optimization algorithm. Finally, the lidar localization results were fused with MEMS IMU data through a state-error Kalman filter to produce smoother and more accurate localization information at a high frequency of 200Hz. The proposed localization system can achieve 3~5 cm in position and 0.05~0.1° in orientation RMS (Root Mean Square) accuracy and outperform previous state-of-the-art systems. The robustness and adaptability have been verified with localization testing data more than 1000 Km in various challenging scenes, including congested urban roads, narrow tunnels, textureless highways, and rain-like harsh weather.

**Keywords:** Lidar localization system; unmanned vehicle; segmentation-based feature extraction; category matching; multi-group-step L-M optimization; map management

## 1. Introduction

Localization is one of the most basic and core technologies of unmanned vehicles. Precise and real-time localization service is needed in many modules, including behavior decision, motion planning and feedback control. At present, a high-cost GNSS (Global Navigation Satellite System) and IMU (Integrated Measurement Unit) integrated navigation system [1] is mostly used in unmanned vehicles. These two complementary systems solve the shortcomings of the low frequency of GNSS and the integration drift of IMU [2–4]. The accuracy of single-point positioning technology is low, about 5~10 m, which cannot meet the needs of unmanned vehicles. The real-time carrier-phase based differential GNSS technology can eliminate satellite orbit and block errors, tropospheric and ionospheric delays to achieve centimeter-level localization accuracy, i.e. RTK (Real Time Kinematic) technology [5,6].

However, the differential signal cannot be received everywhere and the accuracy decreases with the distance from the base station. The GNSS position information is prone to jump because of signal occlusion in urban canyons and the multipath effect in large area of flat and smooth ground or water scenes [7]. In addition, the high-cost GNSS receiver and IMU modules, coupled with the paid real-time differential services, make the cost of this localization solution very high. Considering the above factors, it is necessary to design a more accurate, stable, and economical localization system before the large-scale landing of automatic driving technology.

Lidar (Light Detection and Ranging) can obtain the 3D point cloud of the scene by the multiple rotating laser beams. We can obtain accurate 6DoF (Degree of Freedom) pose in the global coordinate system by matching the online frame with the priori map, it can be used as a better input than RTK in INS (Integrated Navigation System), which can work stably in almost all scenes [8–10]. Hence, how to achieve more accurate and efficient matching is particularly important in a Lidar-based localization system.

Here, we propose a precise and robust segmentation-based Lidar localization system with MEMS (Micro-Electro-Mechanical System) IMU aided. The main contributions of this paper are summarized as follows:

1.  A novel efficient low-level semantic segmentation-based feature extraction algorithm is designed to extract multiple types of stable features from the online frames, including ground, road-curb, edge, and surface. They ensure accurate pose estimation for frame-frame and frame-map matching.
2.  A priori information considered category matching method and a multi-group-step L-M [11] (Levenberg Marquardt) optimization algorithm are proposed, which can avoid most of the mismatching to improve the accuracy, and increase the efficiency by reducing the Jacobian matrix dimension.
3.  An efficient priori map management is presented, the map is stored in tiles with overlap and the local map surrounding the vehicle is dynamically loaded to save the computation resource.
4.  A complex and complete vehicle localization system has been accomplished by integrating all modules reasonably, which can provide high-accuracy and real-time localization service for high-level unmanned vehicles in various challenging scenes and harsh weather.

The remainder of this paper is organized as follows: Section 2 briefly reviews the popular SLAM (Simultaneous Localization and Mapping) system and the Lidar localization technologies fusion with other sensors. Section 3 presents the framework of the proposed segmentation-based Lidar localization system and describes the algorithms used in each module. Section 4 demonstrates the performance qualitatively and quantitatively in various challenging scenes. Finally, we conclude our works and discuss the future research directions in Section 5.

## 2. Related Works

The problem of Lidar localization has been a popular research topic in recent years, which has evolved from SLAM technology [12]. The task of SLAM is to estimate the pose and simultaneously build a point cloud map with vision-based [13–16] or Lidar-based [17–20] methods.

Raul et al. present ORB-SLAM2 [21,22], a complete feature-based visual SLAM system for monocular, stereo, and RGB-D (Color-Depth) cameras, including map reuse, DBoW2-based [23] loop closing, and relocalization capabilities. Gao et al. [24] presents an extension of Direct Spares Odometry [25] to monocular visual SLAM system with loop closure detection and pose-graph optimization. Although vision-based methods have many advantages in loop-closure detection, their sensitivity to illumination and viewpoint change may make such capabilities unreliable if used as the sole navigation sensor.

Lidar can directly obtain fine detailed 3D information of a wide range of scenes even at night. A low-drift and real-time lidar odometry and mapping (LOAM) method is proposed in [17,26], they divide the complex problem into two algorithms, one algorithm performs odometry at

a high-frequency but at low fidelity to estimate velocity, and another runs at a lower order of magnitude frequency for fine matching and registration of the point cloud. LOAM's resulting accuracy is best achieved by a Lidar-only estimation method on the KITTI [27] odometry benchmark site.

On the basis of LOAM, LeGO-LOAM [28] is a lightweight and ground-optimized LOAM for pose estimation of UGVs (Unmanned Ground Vehicle) in complex environments with variable terrain, it is lightweight, as real-time pose estimation and mapping can be achieved on an embedded system. Point cloud segmentation is performed to discard points that may represent unreliable features after ground separation and integrates the ability of loop closures based on ICP (Iterative Closest Point) [29] to correct motion estimation drift. IMLS-SLAM [30] proposes a scan-to-model matching with implicit moving least squares surface representation method, which can achieve a global drift of only 0.69%.

Although there are many excellent back-end optimization and loop-closure detection algorithms, the pose estimation drift will accumulate continuously in long-term large-scale scene with the SLAM scheme and the resulting error will be too large to meet the needs of unmanned vehicles. Therefore, it is necessary to establish a high-precision map in advance and realize accurate localization through matching online frames with the priori map [31–35].

Ryan W [36] exploits the structure in the environment to perform scan match with Gaussian mixture maps, which are a collection of Gaussian mixture over the z-height distribution. They achieve real-time performance by developing a novel branch-and-bound, a multiresolution approach that makes use of rasterized lookup tables of these Gaussian mixtures. Levinson J [37] yields substantial improvements in vehicle localization based on a previous work [31], including higher precision, the ability to learn and improve maps over time, and increased robustness to environment changed and dynamic obstacles. They model the environment as a spatial grid where every cell is represented as its own gaussian distribution over remittance values and then the Bayesian inference is able to preferentially weight parts of the map most likely to be stationary and of consistent angular reflectivity, thereby reducing uncertainty and catastrophic errors.

In the Lidar-only localization system, the output frequency of localization information is too low to meet the needs of high-speed unmanned vehicles, and the trajectory is not smooth enough. Hence, some studies focus on the fusion of more information from multiple type sensors to obtain more reliable localization results [38–40]. An integrated GNSS, INS, and Lidar-SLAM positioning method for highly accurate forest stem mapping is proposed in [39], the heading angles and velocities extracted from GNSS and INS are used to improve the positioning accuracy of the SLAM solution. In [41], they develop a localization system that adaptively uses information from complementary sensors such as RTK, Lidar, and IMU to achieve high localization accuracy and resilience in challenging scenes, Lidar intensity and altitude cues are used instead of 3D geometry to improve the accuracy and robustness, and an error-state Kalman filter is applied to fuse the localization measurements from different sources with novel uncertainty estimation, it can achieve relatively high accuracy at 5~10 cm RMS (Root Mean Square). This method relies on the intensity information of point cloud, but it is not easy to calibrate the Lidar intensity, and each Lidar has its own differences. In addition, its localization accuracy still needs to be improved.

## 3. Methodology

### 3.1. System Overview

The architecture overview and the complete step-by-step algorithm flow of the proposed Lidar localization system are shown in Figure 1 and Algorithm 1. The input of the system includes the raw online cloud, raw MEMS IMU data as well as the prior map dataset, and output real-time accurate 6 DoF pose information.

The system can be divided into five main modules:

(1) Feature Extraction: we organized the raw online point cloud and eliminate the dynamic objects within the road, and then extracted ground, road-curb, surface, and edge features through a series of

proposed efficient low-level semantic segmentation algorithms. The following cloud matching process is based on the extracted features.

(2) Lidar Odometry: frame-frame matching was performed to obtain the ego-motion between two adjacent frames. The result was used as the initial value of frame-map matching in Lidar localization module.

(3) Local Map Load: the local prior feature map is loaded dynamically based on the current vehicle localization information.

(4) Lidar Localization: the current frame was matched with the loaded local map to obtain an accurate global pose and pushback the result into the filter.

(5) Error-state Kalman Filter: we established an error-state Kalman filter that uses Lidar localization result and IMU raw data as measurement input and output high-frequency precise 6 DoF pose. After the filter was initialized, the filter result was used as the initial value of frame-map matching instead of Lidar odometry. The Lidar odometry node lost its meaning and was shutdown to save computing resources.

Furthermore, in order to balance the computing resources and ensure the localization accuracy, each node ran at different frequencies (Figure 1) to form the complete segmentation-based Lidar localization system. The detailed algorithm principle of each modules will be introduced in the following sections.

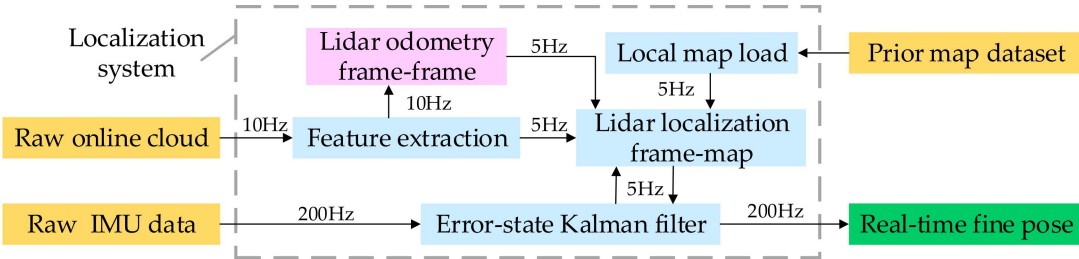

**Figure 1.** Overview of the proposed Lidar localization system architecture. IMU: Integrated Navigation Unit; Lidar: Light Detection and Ranging

---

**Algorithm 1.** Lidar Localization System

---

**Input:** prior map $M$, online point cloud $C$ at 10 Hz, IMU data $I$ at 200 Hz.
**Output:** fine pose $T^a$.
1: establish the error-state Kalman filter $F$, and pushback the IMU data $I_n$ into $F$;
2: **for** frame $C_i$ **do**
3: delete the dynamic object $C_i^d$ within the road;
4: based on low-level semantic segmentation, extract the feature point cloud road $C_i^r$,
    road_curb $C_i^c$, surface $C_i^s$, and edge $C_i^e$ from the remaining point cloud in turn;
5: **if** $F$ initialized **then**
6: calculate the transform $\Delta T$ between the $C_{i-1}$ and $C_i$ from IMU odometry queue;
7: **else**
8: get $\Delta T$ by category matching $C_{i-1}$ with $C_i$;
9: **end if**
10: calculate the rough pose $T_i^r = T_{i-1}^a * \Delta T$;
11: dynamically update and load the local map $M_i$;
12: category match $C_i$ with $M_i$, and use $T_i^r$ as the initial value to get the fine pose $T_i^a$ by
    multi-group-step L-M optimization;
13: pushback $T_i^a$ into $F$;
14: **end for**
15: calculate the final fine pose $T_n^a$ using $F$, and output at 200 Hz;
16: **return** $T_n^a$;

---

*3.2. Lidar Feature Map Management*

The high precision Lidar localization system depends on a priori map. Therefore, it was important to design an effective method to store and load the map of large-scale scenes. Here, our priori map separately stored the extracted multiple types of feature point clouds and the specific feature extraction methods are demonstrated in 3.3. In order to manage a large-scale point cloud map, we extended the two dimensions into three-dimensional tiles to store the map, each tile was identified by $(i, j, k)$, which indicate the index in $x$, $y$, and $z$ directions. In addition, the relationship between the point cloud file path and the tile index was also stored for conveniently finding and loading the specified tiles.

In the localization system, the currently visible area was calculated according to the vehicle position information as well as considering the view angle and measurement range of Lidar. We then loaded the tiles in the area to form the current local map. When loading map data, the K-d tree of the point cloud in each tile was also established to prepare for finding the nearest neighbors of the point cloud matching process in Lidar localization module. To avoid wasting computing resources of re-loading the point cloud and reestablishing the K-d tree, we retained the loaded tiles, only loaded the newly added, and removed the unused when the vehicle position was updated.

The map was stored by tangent tiles in [41], which causes the K-d tree at the tile edge unable to reflect the real nearest neighbor relationship and produces an incorrect result when searching for the nearest points. Here, we proposed an overlapped tile to store the map. Figure 2 is the 2D representation of the relation between tangent tiles (blue) and overlapped tile (green). On the basis of keeping the center of the tile unchanged, the overlapped tile expanded leaf size so that the adjacent tiles are overlapped. Therefore, the K-d tree at the edge (red) of the original tile was ensured to be correct. The overlap does not need to be too large, thus wasting storage space and computing resources, as long as the correctness at the edge (red) can be ensured. Here, we set the overlap as 6 m and the effectiveness had been proved by experience. It should be noted that there was no relationship between overlap and leaf size, and we did not need to adjust overlap when the leaf size was changed in using.

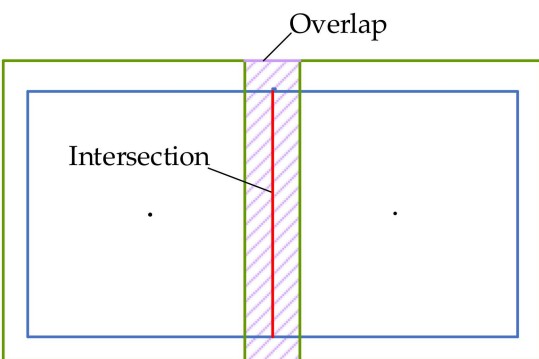

**Figure 2.** The 2D representation of the relation between tangent tiles and overlapped tiles.

Figure 3 shows the scope of the test area in this study, which had a size of about $3 \times 4$ km; (b) is the tiled point cloud map of ground feature and the tiles are given random color. The leaf size of each tile was 50 m and with 6 m overlap between two adjacent tiles. The map accuracy was the premise of the localization system accuracy. We used a total station to measure some obvious feature points on the spot and found the same points in the generated point cloud map to evaluate the map accuracy, which can achieve centimeter level and meet the requirement of Lidar-based localization system.

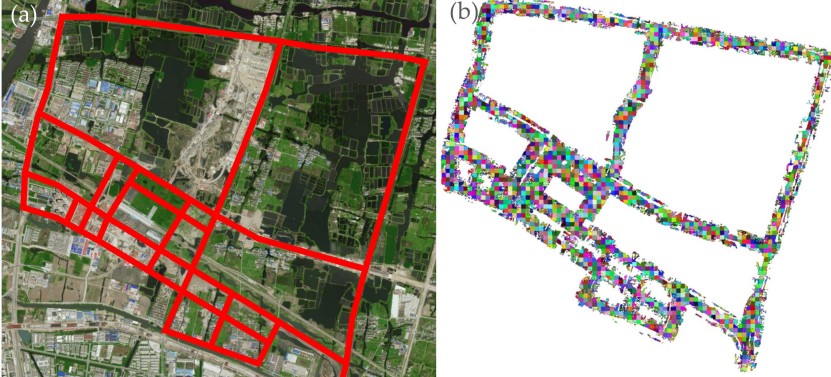

**Figure 3.** The scope of the test area. (**a**) The test roads in satellite image. (**b**) Tiled point cloud map.

### 3.3. Segmentation-Based Feature Extraction

As each frame data contained millions of points, the computation was too large to satisfy real-time processing if all points were used for pose estimation [42–44]. In addition, there were many dynamic objects in the road scene, which were noise in the registration process, and reduced the registration accuracy or even lead to failure [45]. In this study, we first eliminated the dynamic objects within the road by the pre-built RoadROI (Road Region of Interest), which was generated through ground segmentation and region growing. In the following, we separately introduced a series of proposed low-level semantic segmentation-based feature extraction algorithms, including ground, road-curb, surface, and edge.

#### 3.3.1. Ground

Lidar acquired data by continuously rotating laser scans and publishes them as small packets. These packets were collected for the time of the sensor revolution in what we called a frame; the time was 100 ms for Velodyne VLP-32C. Before running our algorithms, we first inertially corrected the frame distortion caused by the vehicle's motion through IMU or odometry information [46].

Then, the point cloud was projected onto a cylinder and expanded into a two-dimensional range image, called an organized point cloud. Each row of the range image represented a laser line, which had the same vertical angle $\theta_v$; each column represented the points of different scan lines at the same azimuth angle $\theta_a$. Therefore, we exploited the clearly defined neighborhood relations directly in the 2D range image without additionally computing resource to build K-d tree, and the 2D range image had more flexibility in use. For example, we easily obtained a point in specified laser line or column and it offered many conveniences for point cloud segmentation. The subsequent segmentation-based feature extraction algorithms were all implemented on the basis of the organized point cloud.

The ground extraction was the primary task of rule-based point cloud segmentation [47], calculating complex normal difference and gradient, then performing region growing to extract ground from a Lidar frame. This occupied too much computing resources, which made it unsuited for such a complex and real-time system. An image of when the laser scanning was on a flat ground surface can be seen in Figure 4. The differences in the $z$ direction was far less than the $x, y$ directions between two adjacent points in the same column (the two points in the purple box of Figure 4) and the former one was close to 0. Considering such a geometric feature, we defined a term $\alpha_i$ to represent the vertical angle between two adjacent points in the same column:

$$\alpha_i = tan^{-1}\left(\frac{\delta_{z,i}^c}{\sqrt{\delta_{x,i}^c * \delta_{x,i}^c + \delta_{y,i}^c * \delta_{y,i}^c}}\right) \tag{1}$$

where $\delta_{x,i}^c$, $\delta_{y,i}^c$ and $\delta_{z,i}^c$ represents the difference in $x, y, z$ direction between two adjacent points in the $c$th column. We traversed all points in $m$ rows below the range image to calculate $\alpha_i$, because only these



laser lines can scan to the ground and $m = 17$ for Velodyne VLP-32C. When $\alpha_i < 2.5°$ and $\delta^c_{z,i} < 0.05$, they are the ground points. Furthermore, the vertical angle plays a more important role than the elevation difference in ground segmentation. The threshold was relatively loose, so the algorithm can still perform well at the transitions between flat and sloped terrain. There was no difference for the algorithm when the vehicle was on a flat surface or slope, because the point cloud is in the Lidar coordinate system, which was relatively parallel to the ground. The segmentation results contain not only road surface but also flat ground, where the flat ground points played the same role as road surface in optimizing pose, so they were also needed in our localization system.

### 3.3.2. Road-Curb

We solved pose using the constraints provided by feature points, while the stable edge and surface features were fewer in textureless scenes, such as a highway where the constraints were fewer, which increased the possibility of matching failure in such scenes. Road-curb was also a kind of structurally stable feature, which provided the constraints of multiple DoF. As shown in Figure 4 [48], road-curb has many spatial features different from the ground surface. For example, the evaluation of road-curb changes sharply while the ground surface points are smooth and continuous. Based on these features, the road-curb detection method is proposed.

We first calculated the theoretical threshold based on the spatial features to separate road-curb from ground surface. $\delta^l_x, \delta^l_y, \delta^l_z$ represents the difference in $x, y, z$ direction between two adjacent points of ground surface in the $l$th laser line (the two points in the red box of Figure 4), and it is defined by

$$\delta^l_x = h_s \cdot cot\theta^l_v \cdot (1 - cos\theta_h) \tag{2}$$

$$\delta^l_y = h_s \cdot cot\theta^l_v \cdot sin\theta_h \tag{3}$$

where $h_s$ is the sensor height, $\theta^l_v$ is the vertical angle of the $l$th laser line, and $\theta_h$ is the angle resolution of the sensor, which is set to 0.2° for Velodyne VLP-32C. If the ground surface satisfies the assumption that it is completely smooth, then $\delta^l_z = 0$. While considering the measurement error and the slight roughness of the ground surface, here we set $\delta^l_z = 0.003m$. All points of the ground surface were then traversed to calculate the difference $\delta^l_{x,i}, \delta^l_{y,i}, \delta^l_{z,i}$ in $x, y, z$ direction between two adjacent points in the $l$th laser line. If $\delta^l_{x,i} > \delta^l_x, \delta^l_{y,i} > \delta^l_y, \delta^l_{z,i} > \delta^l_z$, the points were road-curb candidate points. There was some noise in the candidate points, considering that the road-curb was an approximately straight line, and we iteratively extracted all road-curb lines from the candidate points.

The above algorithm has a hypothesis that the road-curb is approximately parallel to the x- or y- axis of the Lidar coordinate system. The hypothesis was destroyed when the vehicle was turning. Here, we determined the vehicle turning state based on $\theta_{yaw}$ using localization information and then used the accumulated turning angle to invert the point cloud to satisfy the hypothesis. Finally, the road-curb was detected with high precision and recalled in various scenes such as going straight and turning.

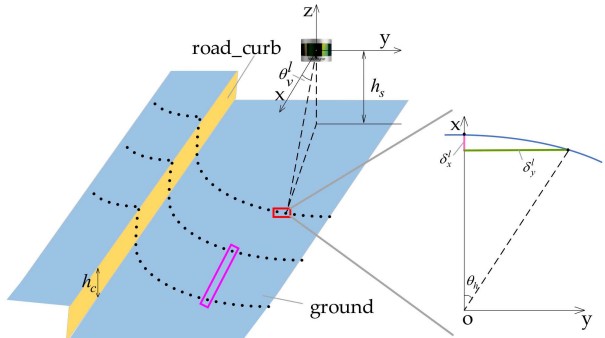

**Figure 4.** Ground and road-curb scene description.

### 3.3.3. Surface

There were also other surface features expect the extracted ground plane, including architecture walls, traffic signs, billboards, etc. According to the scanning characteristics of a horizontally installed Lidar, when the laser line was scanned in a plane, its three-dimensional coordinate values were smooth, continuous, and symmetrical. For a sparse Lidar point cloud of a single frame, when the laser line scanned the distant wall surface, although it was only a single line, but it should be defined as a surface feature instead of edge. In [26], the smoothness of a point is defined, which is used to distinguish surface and edge, but it cannot perform well in a complex urban road environment, and the extraction results are confusing. Here, we define a more concise term $P$ that can achieve better results to describe the planeness of the local surface:

$$P = \left(I_{c-1}^l + I_c^l + I_{c+1}^l\right)/3 - I_c^l \tag{4}$$

where $I_{c-1}^l, I_c^l, I_{c+1}^l$ represent three adjacent points in the $l$th rows of the range image. When $P_x, P_y, P_z$ are all less than the threshold, the point is the candidate surface point. The threshold is set as 0.03m here and it is proportional to the horizontal resolution of LiDAR, which is 0.02° for Velodyne VLP-32C. All the points of expected ground feature points are then traversed to obtain the candidate surface points set, and to eliminate the discontinuous candidate points to get the final surface feature point cloud with high quality.

### 3.3.4. Edge

In the road scene, the stable edge feature included light poles, tree trunks, architecture ridges, etc., all of which are vertical. The consistency of the organized point cloud in the column direction was worse than the row direction. Using an extraction method similar to the surface, which assumes the edge feature points are in the same column of the range image, we could not precisely extract edge features. A more effective approach is presented here.

The edge feature extraction process is shown in Figure 5. First, the remaining points after removing the ground and surface points were clustered, since the traditional Euclidean distance clustering method required a lot of computation resources. Here, we adopted a grid-based method to project point cloud into XOY plane, which is composed by x-axis and y-axis of the Lidar coordinate system. Then, we performed region growing according to the plane distance, which clustered the point cloud quickly and effectively [49]. Each cluster was then fitted with a line using the RANSAC (Random Sample Consensus) method to obtain the points set and normalized line parameters. When the line was approximately parallel to the Z axis, or, in other words, the z in line parameters were close to 1, and x, y were close to 0, the points set was an edge feature point cloud.

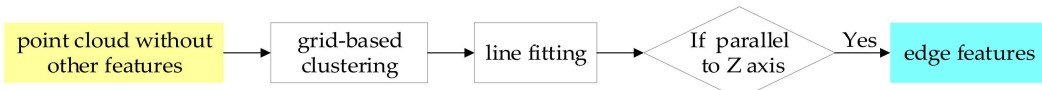

**Figure 5.** Edge feature extraction process.

### 3.4. Priori Information Considered Category Matching

Based on the extracted feature points, frame-frame matching was performed to estimate the sensor ego-motion between two consecutive frames in Lidar odometry module and frame-map matching was performed to obtain the accurate global localization information in the Lidar localization module. The processing of the two modules was almost the same in Sections 3.4 and 3.5, except for a few

differences in the data association part. The details of the processing will be introduced in the following paragraphs.

We extended the matching strategy of point-to-line and point-to-plane in [17], which was more reasonable and robust than the traditional point-to-point for the sparse Lidar data. For the sake of brevity, the detailed procedures of point-to-line and point-to-plane matching can be found in [17]. Here, we propose a priori information considered category matching method to improve the accuracy and efficiency.

(1) Category Matching: In Lidar odometry and Lidar localization modules, we needed to find the corresponding features for points in the current frame $C_i$ from the last frame $C_{i-1}$ and the current dynamically loaded local map $M_i$ respectively. In frame-map matching, the tile index in the map of each point in $C_i$ was calculated and then the k-nearest neighbor points $np^k$ in the correspondence tile are found. There were four categories of feature point clouds extracted by the same method in $C_i$, $C_{i-1}$, and $M_i$, and they are maintained separately. Thus, we only found correspondences in the same category, because the structure of the feature points was stable. Therefore, it could be extracted to the same category feature repeatedly in different frames. For example, for the edge features in $C_i^e$, we found its corresponding points in $C_{i-1}^e$ and $M_i^e$ points set. This method can not only improve the efficiency by reducing the potential candidate points, but also improve the accuracy of data association.

(2) Consider priori information: Before calculating the distance of point-to-line and point-to-plane, the equations of the line and plane need to be calculated. The line and plane equations can be expressed with a direction vector $d(d_x, d_y, d_z)$ or normal vector $n(n_x, n_y, n_z)$ and a point in it, where the point can be obtained by nearest neighbor search, and the key and difficulty is the solution of the vector. In the frame-frame matching of [17] and [28], they first find the two or three nearest points in different laser lines, then use Formula (5) and (6) to calculate the direction and normal vector, respectively.

$$d = np^0 - np^1 \tag{5}$$

$$n = \left(np^0 - np^1\right) \times \left(np^2 - np^1\right) \tag{6}$$

In the frame-map matching, they obtain the vector by performing the PCA (Principal Component Analysis) with the five nearest points and the computation was relatively high. Furthermore, the vector could not precisely express the local spatial geometry because of the sparseness of Lidar data. Here, we considered the priori information of the extracted features. For example, the direction of edge features was vertical and the plane of the ground features parallel to the Lidar coordinate system's XOY plane. Therefore, when using the point-to-line matching method for edge features, we directly assigned the direction vector to $d(0, 0, 1)$ to form a line equation after finding the nearest point. For the simplicity of expression, Table 1 shows the match method, the needed nearest points numbers, the calculation method of direction or normal vector for different feature types in frame-frame, and frame-map matching. The matching methods are the same for frame-frame and frame-map, but the nearest points number and the calculation method of direction or normal vector are different. The computation complexity of the three methods for vector calculation, PCA, direct assignment and formula (5)(6) decrease in turn, but the accuracy is increase. By fully considering the priori information to simplify the solution of direction or normal vector and improving the accuracy of data association, the accuracy and efficiency of pose estimation can be improved.

**Table 1.** The data association method of different types of feature points. PCA: Principal Component Analysis

| Feature Type | Match Method | Nearest Points Num | | Direction/Normal Vector | |
|---|---|---|---|---|---|
| | | Frame-Frame | Frame-Map | Frame-Frame | Frame-Map |
| edge | point-to-line | 1 | 1 | (0,0,1) | (0,0,1) |
| road-curb | point-to-line | 3 | 5 | Formula (5) | PCA |
| ground | point-to-plane | 1 | 1 | (0,0,1) | (0,0,1) |
| surface | point-to-plane | 2 | 5 | Formula (6) | PCA |

### 3.5. Multi-Group-Step L-M Optimization

The distance of all correspondences of the feature point cloud was calculated by point-to-line and point-to-plane distance formulas. Therefore, the Lidar motion was recovered by minimizing the overall distances. The formula for optimizing the pose using the gradient-descent-based L-M method is deduced in detail in [26], where optimize 6DoF $\{t_x, t_y, t_z, \theta_{roll}, \theta_{pitch}, \theta_{yaw}\}$ with all the points together. This results in a higher dimension Jacobian matrix $j$ and an increase in the computational resource consumption. In addition, the ground points have no benefit for optimization $\{t_x, t_y, \theta_{yaw}\}$, or even reduce accuracy.

Due to the different spatial characteristics of each type of feature points, there was a significant difference in their ability to constrain the variables in the pose. For example, the ground points have a strong constrain on $\{t_z, \theta_{roll}, \theta_{pitch}\}$, but were invalid for other variables. Furthermore, the edge points were good at constraining $\{t_x, t_y\}$, but they do not make sense for $\{t_z\}$. Here, we propose a multi-group-step optimization method; Figure 6 shows the pipeline. It was successively optimized by four steps of edge, road-curb, ground, and surface, wherein each step optimizes different variables (shown in the box bottom) that depended on the feature type, and the optimized result of each step was used as the initial value of the next step, better initial values reduced the likelihood of falling into a local optimum. After a four-step optimization, we can obtain a more accurate pose from the initial coarse pose. In addition, since the Jacobian matrix dimension in each step was relatively lower, the computation time was reduced. It should be noted that the optimization was orderly, the edge feature with the least constraints was used for optimization first, and the surface feature with the most constraints was used for optimization last; experiments verify that the best results can be obtained in this order. The Multi-group-step L-M optimization method can not only reduce the computing resource but also improve the accuracy of pose estimation.

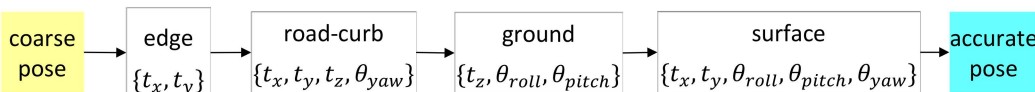

**Figure 6.** The pipeline of Multi-group-step L-M optimization.

### 3.6. Lidar and IMU Fusion

The output frequency of the pure Lidar localization system only reached 10 Hz at most, which could not meet the needs of unmanned vehicles running at high speed. IMU can produce a high frequency triaxial acceleration and rotation rate and it is sufficiently accurate to provide robust state estimations between Lidar measurements. IMU and Lidar are two complementary sensors and we can obtain more accurate and smooth real-time localization results after fusing their information.

Here, we used an error-state Kalman filter fusion framework to fuse the Lidar and MEMS IMU measurements. The error-state was the difference between the estimated state and truth state, including position, orientation, velocity, gyroscopes biases, and accelerometers biases. With the MEMS IMU data, the SINS (strap-down inertial navigation system) was used to predict the position, orientation, and velocity as a prediction model in the Kalman filter propagation phase by integrating the specific force measured by the accelerometer and the rotation rate measured by the gyroscope. With Lidar pose measurements that estimated by Lidar frame-map matching, we update the error-state Kalman Filter's state and then used the error-state to correct the SINS state as an update model in the Kalman filter. The fusion framework is similar to [41] and the input and output difference are shown in Table 2. The proposed system's input is Lidar localization results and MEMS IMU data, dose not rely on RTK. The output includes 6DoF pose $\{t_x, t_y, t_z, \theta_{roll}, \theta_{pitch}, \theta_{yaw}\}$, velocity, accelerometers, and gyroscopes bias. After the fusion of Lidar localization information and MEMS IMU data, the system produced

a smoother, high-frequency and high-accuracy 6DoF pose, providing localization service for other modules of unmanned vehicles.

**Table 2.** The filter input and output difference between [41] and ours. RTK: Real Time Kinematic; DoF: Degree of Freedom.

| Method | Input | Output |
|--------|-------|--------|
| [41] | RTK, Lidar Localization, IMU | 2D position, altitude, velocity, accelerometers and gyroscopes bias |
| Ours | Lidar Localization, IMU | 6DoF pose, velocity, accelerometers and gyroscopes bias |

## 4. Experiment Results and Discussions

### 4.1. Hardware System and Evaluation Method

The proposed system was validated on a computer with a 3.6GHz i7-8700 CPU processor, 16GB memory, and GPU was not used in the experiment. A Velodyne VLP-32C was horizontally installed on the car roof to collect LiDAR data. The VLP-32C measurement range was up to 180 m with an accuracy of ±3 cm, it had 32 laser channels with a 30° vertical FOV (Field of View) and 360° horizontal FOV, and it could obtain data up to 10 Hz. The MEMS IMU(ADIS16465) was installed under LiDAR and it included a triaxial digital gyroscope and triaxial digital accelerometer. In addition, the unmanned vehicle was equipped with a high-cost and high-precision RTK/IMU INS(XW-GI7660) for evaluation. The INS achieved 0.02 m in position and 0.01° in orientation RMS accuracy.

In order to evaluate the localization accuracy quantificationally, the ground truth of the vehicle motion trajectories should be established. We have designed two methods for different scenes, which with good and bad GNSS signals respectively. **Online evaluation mode:** The ground truth was generated through the INS, but it could only be used in an open scene with a good GNSS signal and the accuracy declined rapidly when entering the areas with poor GNSS signal such as urban canyons. **Offline evaluation mode:** The base station was set up at the control point to post-process the original RTK/IMU integrated navigation data and some Lidar SLAM technologies were added, including scan-match, loop closure, and global pose-graph optimization, we finally got more accurate trajectories as the ground truth. Actually, that was also the pipeline of Lidar mapping.

We calculated the absolute 6 DoF errors, where the units were expressed in meters for translation and in degrees for rotation. Then we statistically calculated the $1\sigma$, $2\sigma$, and $3\sigma$ of a data section, which indicated 68.26%, 95.4%, and 99.73% of the errors was less than the value respectively, and the $1\sigma$ was the RMS, and each data section was about 5 km.

The errors have been transformed from the world coordinate system to car coordinate system, in which x and y denote the vehicle forward and transverse direction respectively. The localization system performance can be expressed more clearly.

### 4.2. Segmentation-Based Feature Extraction Result Analysis

The feature extraction module plays a vital role in the proposed localization system. Hence, we first demonstrated the proposed multiple segmentation-based feature extraction algorithms in various scenes before analyzing the localization performance. If we quantificationally evaluated the feature extraction algorithms' performance by calculating the detection rate of each type of feature, a dataset with detailed manual labeling was needed, which required substantial of labor and time. At present, our data were not labeled and there was no such public data set available for use. Therefore, we qualitatively evaluated the segmentation algorithms using some representative scenes. In Figures 7–11, the gray points are the original point cloud of a frame and the blue, red, green, and purple points were the extracted ground, road-curb, surface, and edge features respectively.

As shown in the figures, most of the features can be detected, and the detail is explained in the following paragraphs.

Figure 7 shows the ground segmentation results in different scenes. Figure 7a is a scene where vehicles pass through the crossroads, and the ground in all directions can be completely detected. For some challenging cases, for example, Figure 7b representing a congested urban road with severe occlusion, Figure 7c representing a highway ramp with slope, the algorithm is robust, and can still achieve the same performance as the normal scene. The detection results include not only the road surface but also the flat ground.

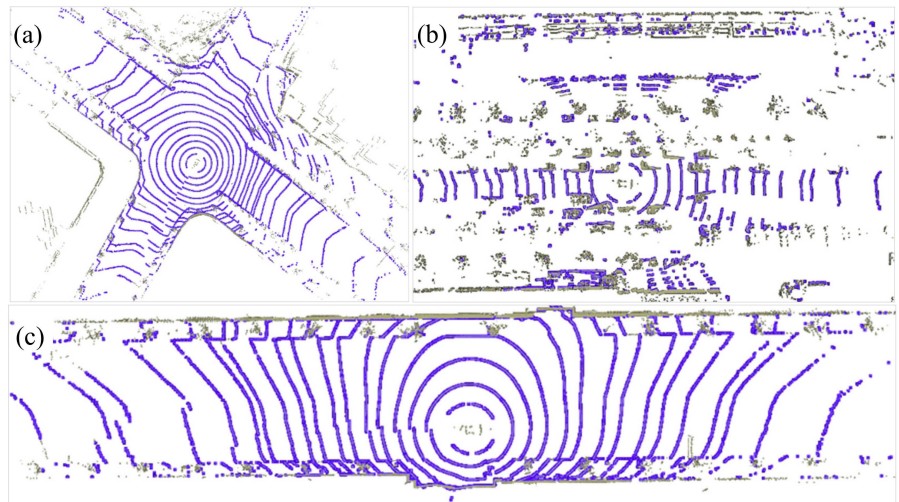

**Figure 7.** The ground segmentation results in various scenes. (**a**) Crossroads scene; (**b**) congested urban road scene; (**c**) highway ramp with slope scene.

Figure 8 shows the road-curb segmentation result in some representative scenes with different difficulty levels to demonstrate the wide adaptability of the proposed algorithm. Figure 8a is the simple straight road scene, where the multiple road-curb line on both sides of the road can be detected steadily. Figure 8b shows the vehicle going straight through the intersection, the road-curb of the two roads parallel and perpendicular to the vehicle travel direction can all be detected. For the most challenging turn scenes, the algorithm can still work by adding the reverse rotation point cloud processing, Figure 8c,d are left-turn and right-turn cases respectively.

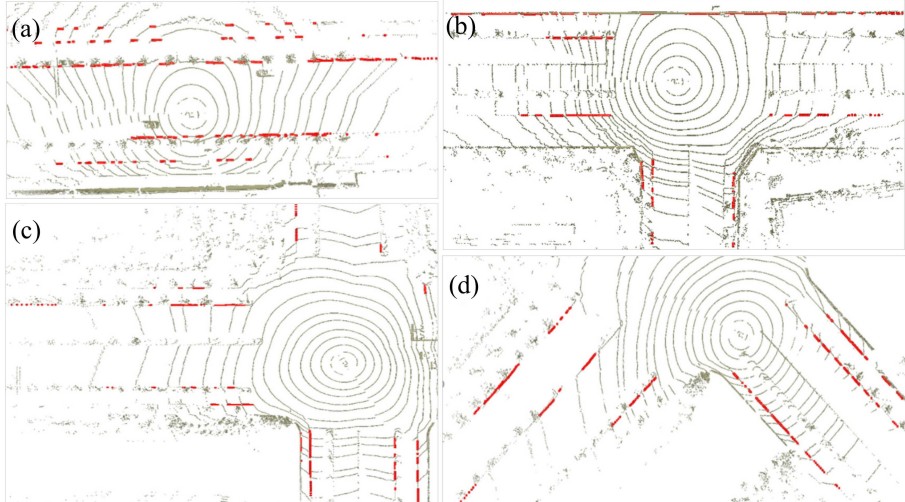

**Figure 8.** The road-curb segmentation results in various scenes. (**a**) Straight scene; (**b**) straight through the intersection scene; (**c**) left-turn scene; (**d**) right-turn scene.

Figure 9 shows the surface segmentation results, including architecture walls, traffic signs, billboards, etc. For sparse multi-channel Lidar data, these high and distant horizontal lines should still be defined as surface features. In addition, it can eliminate most of the scattered leaves and other unreliable points, which should be considered as noisy, as they are unlikely to be seen in two consecutive frames.

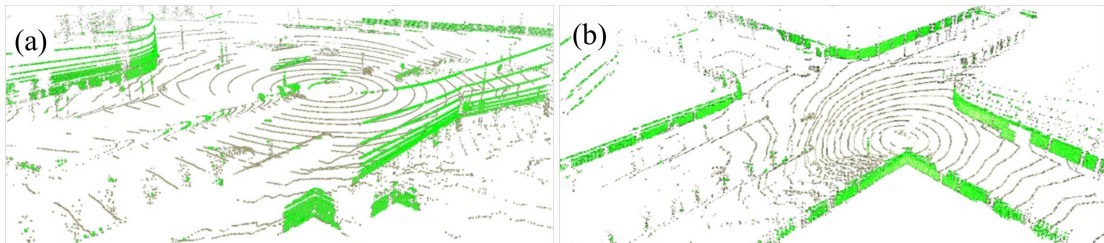

**Figure 9.** The surface segmentation results in various scenes; (**a**) and (**b**) are two different scenes.

The edge segmentation results are shown in Figure 10. It can extract almost all the rod-like objects at a very long range, such as lamp poles, traffic poles, tree trunks, and the vertical lines of architecture. There may be a few false negative points, for example, multiple columns laser points scan on a lamp pole that is very close to the vehicle, the lamp pole will be detected as surface instead of edge feature. Such a case is rare, and compared with [26] and [28], our precision and recall are far higher, and their extracted edge and surface features are confusing.

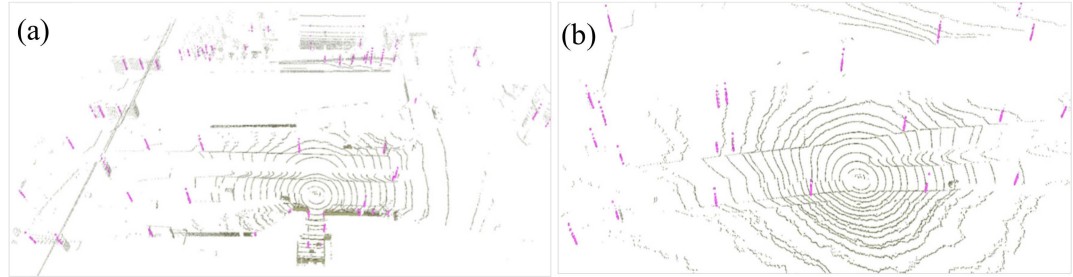

**Figure 10.** The edge segmentation results in various scenes; (**a**) and (**b**) are two different scenes.

We propose a series of efficient low-level semantic segmentation algorithms for sparse Lidar point cloud by making full use of the spatial geometric features. They are not only robust for all kinds of challenging scenes but also very lightweight with low time and space complexity, the time consumption of each frame is only about 35ms. The algorithms can work stably on online frames. Figure 11 shows the performance in three scenes, it can be seen that the algorithms can eliminate dynamic objects, remove the cluttered tree leaves points from the raw point cloud, and segment them into four structured semantic point clouds, including ground, road-curb, edge, and surface. The segmentation results are downsampled by the voxel grid method, and then fewer feature points are used for matching.

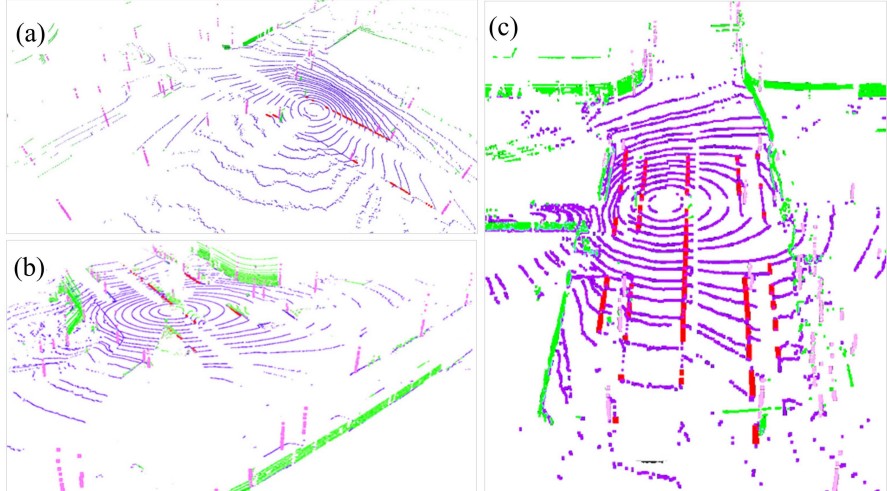

**Figure 11.** The efficient low-level semantic segmentation-based feature extraction results; (**a**), (**b**), and (**c**) are three different scenes.

### 4.3. Qualitative and Quantitative Analysis of Localization Accuracy

Our Lidar-based localization system was deployed in several unmanned vehicles and conducted daily localization tests to verify the stability of the system. The main test area is shown in Figure 3 and the total test mileage is more than 1000 Km. Here, we selected three typical sections of data of scenes with varying difficulty to evaluate the performance qualitatively and quantitatively. S1 is a typical urban road scene, S2 is a crowed urban road scene in rainy day, and S3 is a textless highway scene.

The difficulty of S2 was that a large number of dynamic vehicles point cloud in crowded urban roads were noise for the localization and block the view of the Lidar. In addition, harsh weather, such as rain and fog, had similar effects on Lidar data, which reduced the Lidar measurement range and created some noisy points; this phenomenon was more serious in rainy days. Therefore, we selected a section data of rainy day with worse quality for comparison. While the system first removed the most dynamic objects and the segmentation-based feature extraction algorithms had a strong suppression effect on noise, the limited point cloud information was fully used to determine a high-precision pose.

In S3, there were almost no stable surface features in the highway scene, such as Figure 10b, the constraints were too few, causing potential fail in pose estimation. The edge and road-curb could be extracted to ensure the number of constrains and the system work stably with the proposed matching and optimization algorithms.

The segmentation-based feature point extraction method (Section 3.3) extracted the most structurally stable points in the scene and provided enough basis for point cloud matching. The proposed priori information considered category matching method (Section 3.4) added the filter of category matching, which effectively reduced mismatching and improved the correctness of data association. Furthermore, in the proposed multi-group-step L-M optimization method (Section 3.5), we fully considered the characteristics of each features and analyze which variables of pose were constrained by them, and we divided the six pose six variables into groups and solved them step by step. All of these improvements ensured that the system worked steadily and precisely. Each online frame matched well with the priori map and the estimated trajectory closely coincides with the ground truth. The computational complexity was small enough to meet the need for real-time processing and no frame skipping occurs.

We show the quantitative results in Table 3, where the units are expressed in meters for translation and in degrees for rotation, and the evaluation methods of which have been expounded in 4.1. Reasonably, the translation and rotation errors in all directions of S1 were better than S2 and S3, but they were still on the same level. In addition, most errors of S2 were slightly less than S3. This shows that textless is more challenging than noise for the proposed Lidar localization system, which was due to

the strong suppression of noise by the proposed feature extraction method. In order to calculate a more accurate pose, we needed to optimize to make full use of the limited information in the textless scene to calculate a more accurate pose. The RMS errors of translation achieved 3~5 cm and the maximum errors of $t_x, t_y$ were less than 15 cm. The RMS errors of rotation are less than 0.1° and mainly in $\theta_{yaw}$, and even though $\theta_{yaw}$ was larger than $\theta_{roll}$ and $\theta_{pitch}$, it was small enough to be used. In fact, the changes in $\theta_{roll}$ and $\theta_{pitch}$ were very small when the vehicle travels. In the existing reports [41], is the most complete and accurate state-of-the-art Lidar-based localization system, where the $t_x, t_y$ achieved 5~10 cm RMS accuracy and $t_x$ is about 10 cm RMS. Our system outperformed the previous state-of-the-art systems and the localization error was reduced by half.

**Table 3.** Pose errors in the three different scenes.

| Scene | X | | | Y | | | Z | | |
|---|---|---|---|---|---|---|---|---|---|
| | 1σ | 2σ | 3σ | 1σ | 2σ | 3σ | 1σ | 2σ | 3σ |
| S1 | 0.034 | 0.065 | 0.090 | 0.030 | 0.064 | 0.100 | 0.042 | 0.124 | 0.201 |
| S2 | 0.040 | 0.099 | 0.132 | 0.035 | 0.067 | 0.118 | 0.048 | 0.127 | 0.218 |
| S3 | 0.048 | 0.080 | 0.125 | 0.040 | 0.069 | 0.100 | 0.046 | 0.134 | 0.211 |
| Scene | Roll | | | Pitch | | | Yaw | | |
| | 1σ | 2σ | 3σ | 1σ | 2σ | 3σ | 1σ | 2σ | 3σ |
| S1 | 0.018 | 0.037 | 0.066 | 0.019 | 0.034 | 0.042 | 0.083 | 0.173 | 0.208 |
| S2 | 0.022 | 0.049 | 0.073 | 0.019 | 0.041 | 0.082 | 0.080 | 0.186 | 0.275 |
| S3 | 0.057 | 0.089 | 0.100 | 0.018 | 0.034 | 0.042 | 0.100 | 0.201 | 0.227 |

Figure 12 is the localization error curves of a data piece with about 2600 s, which include almost all kinds of scenes, such as typical urban road, congested urban roads, narrow tunnels, and textureless highways. Figure 12a,b represent translation and rotation errors respectively. The fluctuation of the $t_z$ and $\theta_{yaw}$ are larger than $t_x, t_y$ and $\theta_{roll}, \theta_{pitch}$, while they are within the accuracy requirements of planning, control and other modules in unmanned vehicles. In fact, the $t_z$ error also confused some other factor that the ground truth in $t_z$ is not accurate enough as the RTK accuracy in the $z$ direction is lower than $x$ and $y$.

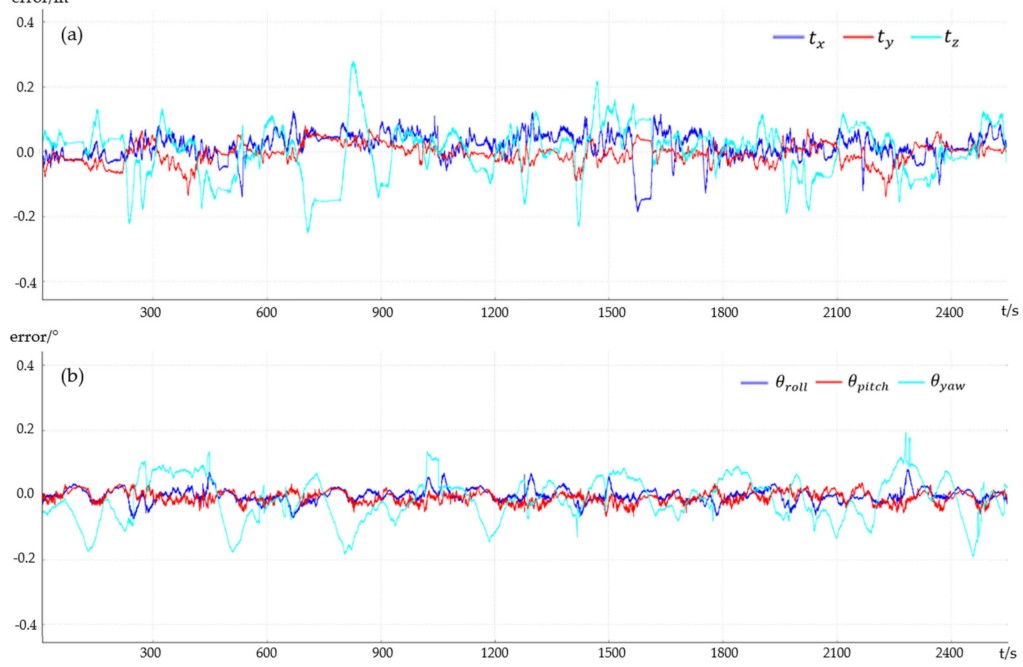

**Figure 12.** Pose error curves. (**a**) Position error curves; (**b**) orientation error curves.

## 5. Conclusions

We have proposed a complex and complete segmentation-based Lidar localization system, designed for high level autonomous driving applications, which can not only reduce the cost of localization, but also improve the accuracy and stability. The implementation of the precise and robust real-time localization system benefits from a series of optimization improvements, including the prior map storage management and dynamic loading method, the efficient low-level semantic segmentation-based multiple type feature point clouds extraction algorithm, the priori information considered category matching data association method, the multi-group-step L-M iterative optimization algorithm, the fusion with MEMS IMU data through a state-error Kalman filter, and the reasonable and effective coupling of multiple modules in the system framework.

Our localization system can produce a stable and precise 6DoF pose at 200Hz, achieve 3~5 cm in position and 0.05~0.1° in orientation RMS accuracy; the accuracy outperforms previous state-of-the-art systems. These results are sufficient to meet the automatic driving requirements in control and planning modules. The system can achieve almost identical performance in various challenging scenes, including congested urban roads, narrow tunnels, and textureless highways. In rain, fog, and other harsh weather, even though the Lidar range decrease and the noise increase, the system can still work normally. Our system has been deployed in multiple unmanned vehicles and large-scale test verifications have been conducted every day. The robustness and adaptability have been verified with more than 1000 Km localization testing data.

As the proposed Lidar localization system relies on the prior map dataset, if there are many large modifications in the scene, such as the architecture demolition or construction, the accuracy will decrease or the localization may even fail. Hence, methods to update the map dataset automatically, efficiently and cheaply will be one of the researches focus in future work. In addition, using the model-to-model data association method instead of the used point-to-line or point-to-plane methods can further reduce the computation complexity of the system, which is also a research topic of future interests.

**Author Contributions:** Conceptualization, H.L., Q.Y. and H.W.; Data curation, H.L., L.C. and J.Y.; Formal analysis, H.L. and H.W.; Funding acquisition, Q.Y.; Investigation, H.L., Q.Y. and H.W.; Methodology, Q.Y., H.L. and H.W.; Project administration, Q.Y.; Resources, H.L., L.C. and J.Y.; Software, H.L. and H.W.; Supervision, H.L.; Validation, H.L., H.W., L.C. and J.Y.; Visualization, H.L.; Writing—original draft, H.L.; Writing—review & editing, H.L. and Q.Y., J.Y.

**Funding:** This research was funded by the National Natural Science Foundation of China under grant number 41771480.

**Acknowledgments:** The authors would like to gratefully acknowledge Beijing Momenta Technology Company Limited who provided us the data support.

**Conflicts of Interest:** The authors declare no conflict of interest.

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
