# Peer review of "A Precise and Robust Segmentation-Based Lidar Localization System for Automated Urban Driving"

_remotesensing, doi:10.3390/rs11111348_

Round 1

Reviewer 1 Report

Very interesting topic. Unmanned vehicles are a complicated interdisciplinary problem, and the ability of vehicles to locate their position in a road environment is perhaps the most important problem to be solved. The material presented by the authors in my opinion is a step on the way to an effective solution to this problem.

Author Response

       Very interesting topic. Unmanned vehicles are a complicated interdisciplinary problem, and the ability of vehicles to locate their position in a road environment is perhaps the most important problem to be solved. The material presented by the authors in my opinion is a step on the way to an effective solution to this problem.

       Thanks for your affirmation of our studies. The manuscript has been revised and improved according to the valuable comments provided by other reviewers. We have carefully checked the grammar and spelling mistakes and used a professional English editing service.

Reviewer 2 Report

This study presents a lidar based localization system based on segmentation algorithms for autonomous driving. Solid works can be found in the article. The topic of this study is interesting as part of this state-of-the-art research thrust and fits for the scope of the Remote Sensing journal.  However, the paper cannot be accepted to be published with current status, because the research work itself is presented with some critical information is missing in this paper. 

The paper should be MAJOR REVISION OR REJECTED but encourage to re-submit when the missing critical information is added.  

The several major issues that the authors should express more clearly to convince the reviewers and also the readers

1.       in Section 4.1 Authors declare that the ground truth is generated  by  online/offline evaluation mode, with what kind of high-cost  and high-accuracy RTK/IMU system, with what kind of laser scanner? (Velodyne VLP-32C or other models, it become a problematic conclusion if the map is generated with the same laser scanner)  how to guarantee the accuracy of the priori map in offline mode, especially in a more than 1000 km dataset? The map accuracy of the prior map is the premise of system accuracy, otherwise, the declared accuracy is skeptical.

2.       The fusion frame should be introduced, it is not reader or reviews’ obligation to read another paper before fully understood this paper.

3.       The authors declared that the processing time for each frame is 35 ms. The configuration of the processed computer should be clearly described to peer review purpose.

4.       Figure 6-9 is not necessary since Figure 10 has present overall performance.  However, the detailed detection rate of each type of features should be discussed (due to the occlusion, it is impossible to detect all road furniture).  The authors should explain how the system can survive in high undetected cases and prove it and how the frame-map matching in such insufficient information cases rather than say “steadily without any failure”

Besides, there are still some minor grammar errors can be found in context, proofreading by a native speak is strongly recommended. Please check the attached pdf for details. 

Author Response

This study presents a lidar based localization system based on segmentation algorithms for autonomous driving. Solid works can be found in the article. The topic of this study is interesting as part of this state-of-the-art research thrust and fits for the scope of the Remote Sensing journal. However, the paper cannot be accepted to be published with current status, because the research work itself is presented with some critical information is missing in this paper. The paper should be MAJOR REVISION OR REJECTED but encourage to re-submit when the missing critical information is added. The several major issues that the authors should express more clearly to convince the reviewers and also the readers.

Thanks for your constructive comments and suggestions to improve the quality of the manuscript. Those comments are all valuable and helpful for revising and improving our manuscript. We have studied comments carefully and have made correction which we hope meet with approval. Revised portion are marked in red in the manuscript. The point-by-point explanation to the comments are as flowing:

Point 1: In Section 4.1 Authors declare that the ground truth is generated by online/offline evaluation mode, with what kind of high-cost and high-accuracy RTK/IMU system, with what kind of laser scanner? (Velodyne VLP-32C or other models, it become a problematic conclusion if the map is generated with the same laser scanner) how to guarantee the accuracy of the priori map in offline mode, especially in a more than 1000 km dataset? The map accuracy of the prior map is the premise of system accuracy, otherwise, the declared accuracy is skeptical.

Response 1: Thanks for your suggestion. XW-GI7660 is the used RTK/IMU INS for generating ground truth, it can achieve 0.02m in position and 0.01° in orientation RMS accuracy. A Velodyne VLP-32C is installed horizontally on our unmanned vehicle for mapping and localization. The detailed introduction of the hardware system is added to section 4.1, line 386~393.

Maybe because our expression is not clear enough, result in you have a misunderstand that our conclusion is problematic when the mapping and localization process using the same sensor. Now let’s give a detailed introduction to the relationship among ground truth, priori map and localization to answer your doubts.

First, in order to evaluate the localization accuracy, the ground truth we need is the vehicle’s motion trajectory, not including the map. For the online evaluation mode when GNSS signal is good, we directly use the original trajectory output by INS as the ground truth. When the GNSS signal is not good, the accuracy of the INS original trajectory can no longer meet the requirement as the ground truth. Therefor, we post-processing the original INS data to improve the trajectory accuracy, the commonly used post-processing software is IE (Inertial Explorer). In addition, some Lidar SLAM methods are also added to further improve the trajectory accuracy, such as scan-match, loop closure detection and global pose-graph optimization. After these processing, we can get a more accurate trajectory to be used as the ground truth for localization accuracy evaluation.

The process of mapping can be divided into two parts. The first part is to obtain a globally consistent trajectory, which is similar to the process of generating the trajectory ground truth of off-line mode. The second part is to extract feature points from the original point cloud data, and use the method introduced in section 3.2 to store and manage the map. We used a total station to measure some obvious feature points on the spot, and found the same points in the generated point cloud map to evaluate the map accuracy, which can achieve centimeter level. We have a main test area, as shown in figure3, we first create a map of this area, and then conduct daily localization tests in this area based on the pre-build map. The 1000Km mentioned in the manuscript is the mileage of localization testing, not the mileage of map. We don’t need to generate map by every day’s localization data, the map only needs to be generated one time at the beginning, and then the generated map is used for daily localization test. That is to say, the data used to evaluate the localization accuracy is not the data used to generate the map, so the conclusion is not problematic.

As you said, the map accuracy of the prior map is the premise of system accuracy, there are a small part of our localization error is due to the map error. If we have a map with no error, our localization error is likely to smaller, but we cannot get such a map without any error. And we have added some essential explanation into the manuscript, it can be seen in line 189~192, 463~465, 533.

Point 2: The fusion frame should be introduced, it is not reader or reviews’ obligation to read another paper before fully understood this paper.

Response 2: Thanks for your suggestion. We have added the introduction of our fusion framework and rewritten this paragraph, and it can also be seen in line 369~382.

Here, we use an error-state Kalman filter fusion framework to fuse the Lidar and MEMS IMU measurements. The error-state is the difference between the estimated state and truth state, including position, orientation, velocity, gyroscopes biases and accelerometers biases. With the MEMS IMU data, the SINS (strap-down inertial navigation system) is used to predict the position, orientation and velocity as a prediction model in the Kalman filter propagation phase by integrating the specific force measured by the accelerometer and the rotation rate measured by the gyroscope. With Lidar pose measurements which estimated by Lidar frame-map matching, we update the error-state Kalman Filter’s state and then use the error-state to correct the SINS state as an update model in the Kalman filter. The fusion framework is similar to [40], and the input and output difference are shown in Table 2. Our system’s input is Lidar localization results and MEMS IMU data, don’t rely on RTK. The output includes 6DoF pose , velocity, accelerometers and gyroscopes bias. After the fusion of Lidar localization information and MEMS IMU data, the system outputs more smooth, high-frequency and high-accuracy 6DoF pose, provides localization service for other modules of unmanned vehicle.

Point 3: The authors declared that the processing time for each frame is 35ms. The configuration of the processed computer should be clearly described to peer review purpose.

Response 3: Thanks for your suggestion. The proposed system is validated on a computer with a 3.6GHz i7-8700 CPU processor, 16GB memory, and GPU is not used in the experiment. We have added the missing critical information into line 386~387. In addition, we also add the information of other hardware used in the system, including Lidar, MEMS IMU and INS, it can be seen in line 386~393.

Point 4: Figure 6-9 is not necessary since Figure 10 has present overall performance.  However, the detailed detection rate of each type of features should be discussed (due to the occlusion, it is impossible to detect all road furniture). The authors should explain how the system can survive in high undetected cases and prove it and how the frame-map matching in such insufficient information cases rather than say “steadily without any failure”.

Response 4: Thanks for your suggestion. Figure 6~9 specifically illustrate the performance of various segmentation algorithms in representative scenes with difference difficulty levels. For example, we select intersection, crowded urban road and ramp scenes for ground segmentation, and select straight, going straight through the intersection and turning at the intersection scenes for road-curb segmentation. While, figure 10, in which various features are put together, shows the overall performance and cannot well express the robustness of various segmentation algorithms in different scenes.

If we want to calculated the detection rate of each type of features, we need to manually label all points in all frame data, and then make point-by-point statistics on the segmentation results to calculate the accuracy and recall rate of each type of features. While, it is a very huge project to manually label the original point cloud in such detail, which required a lot of labor and time. At present, our data are not labeled, and there is no such public data set available for use. Therefore, we qualitatively evaluate the segmentation algorithms by some representative scenes. Of course, this is a very good suggestion, if there is a public data set labeled in detail, it is very beneficial to evaluate the algorithm performance for the research related to point cloud segmentation. At present, most of the existing datasets are only concerned with vehicle labeling, because vehicle perception is the most important task for unmanned vehicle system. With the maturity of vehicle perception schemes, more and more scholars will pay attention to other perception tasks, such as road furniture.

To solve the vehicle’s pose, it is no necessary to detect all features of the scene. We can solve out an accurate pose if we know more than four pairs of homonymous matching points in two frames data that are completely correct and have no errors. In fact, there are some measurement errors, matching errors, line or plane parameters fitting errors, etc. More point pairs can provide more constraints to eliminate the influence of errors by adjustment. In addition to the number of constrains, the accuracy the solved pose is also affected by other factors, such as the correctness of data association. The proposed priori information considered category matching method (section 3.4) adds the filter of category matching, which can effectively reduce the mismatching and improve the correctness of data association. What’s more, how to use the associated data can also affect the accuracy of the solved pose. In the proposed multi-group-step L-M optimization method (section 3.5), we fully consider the characteristics of each features and analyze which variables of pose are constrained by them, and we divide pose’s six variables into groups and solve them step by step. They are all the reasons why our system can obtain stable and accurate results. We have added the explanation into the manuscript in line 413~420, 480~487.

Point 5: Besides, there are still some minor grammar errors can be found in context, proofreading by a native speak is strongly recommended. Please check the attached pdf for details.

Response 5: Thanks for your suggestion. We have carefully checked the grammar and spelling mistakes and used a professional English editing service.

Reviewer 3 Report

Overall the English editing needs to be revisited, a fair amount of mistakes were detected in grammar.

Figures 6-10 are hard to read.  Perhaps a different color scheme would be more clear.

The abstract and conclusion reference 1000km of data, but this has no context of what that data is.  More clarification on the kind of data and how it relates to the experimental procedure is desired.

Section 2 would benefit from a more thorough discussion of the relevant papers, it's not clear on the relevance of the identified papers mentioned.  There is a small discussion on what the papers present but no discussion on how it relates to the authors work.

[98] "ICP" is an unidentified acronym.

[118] "fusion" should probably be "combining"

[Algorithm 1: line 5: "F" is undefined.

[176] what do you mean by "leaf size"?

[Section 3.3.1.] needs a figure/illustration to help present the discussion topic.

[197] please identify the type of Velodyne sensor, there is more than one available on the market.

[212] please rewrite the equation for clarity.

[221-222] this sentence is confusing to read.

[234] this new line should probably be part of the previous paragraph.

[247] what do you mean by "paperback"?

[256] please discuss why the threshold is set to 0.03m.

[Section 3.3.4.] please add an illustration/figure to help present the topic.

[268] "XOY" is undefined.

[270] "RANSAC" is undefined.

[275] what do you mean by "lego-motion"?

[278] the sentence ends with "next", it seems to not be finished properly, please rewrite.

[315] Table 1 might do well with an extra column describing a computation metric because that is part of your discussion in the paper.  

[316] what does "L-M" mean?

[338] Figure 5 can be redone to be more clear and straight forward in presenting the material.  It's a little hard to absorb as is.

[461] you identify that this work is "designed for high-level autonomous driving applications" but I didn't get that emphasis in the rest of the paper.  

[469] is "200Hz" the state of the art?  I didn't feel that it was based on the background in the paper.

[470] this half of the sentence is unclear and hard to read.

Additionally, it looks like the author put a lot of work into this effort but the paper didn't convey the sense of what was innovative and important about the work.  

Author Response

Thanks for your hard work on our manuscript. Your constructive comments and suggestions are very valuable for improving the manuscript. The revised manuscript has been uploaded. Here is the point-by-point response for your comments.

Point 1: Overall the English editing needs to be revisited, a fair amount of mistakes were detected in grammar.

Response 1: We are so sorry for the poor English expression. We have carefully checked the grammar and spelling mistakes and used a professional English editing service.

Point 2: Figures 6-10 are hard to read. Perhaps a different color scheme would be more clear.

Response 2: Thanks for your suggestion. We have changed the background to white, they are clearer to read. It can be seen in the revised manuscript.

Point 3: The abstract and conclusion reference 1000km of data, but this has no context of what that data is. More clarification on the kind of data and how it relates to the experimental procedure is desired.

Response 3: Thanks for your suggestion. 1000Km is the mileage of dataset that used for localization test. The dataset contains the Lidar, MEMS IMU data used for our proposed Lidar-based localization system and INS data used for evaluation. We have a main test area as shown in figure 2, our Lidar-based localization system has been deployed on several unmanned vehicles. Daily localization tests conducted in this area to verify the stability of the localization system, and the total test mileage is more than 1000Km. We have added the explanation into the manuscript in line 463~465.

Point 4: Section 2 would benefit from a more thorough discussion of the relevant papers, it's not clear on the relevance of the identified papers mentioned. There is a small discussion on what the papers present but no discussion on how it relates to the authors work.

Response 4: Thanks for your suggestion. This manuscript discusses the Lidar-based localization system. Lidar localization evolves from SLAM technology. In section 2, we first briefly introduce the problems existing in visual SLAM, and the advantages of Lidar SLAM. Then introduced the principles of some well performed Lidar SLAM algorithms. Next, due to the problem of error accumulation in SLAM system, even if excellent graph-optimization and loop closure detection are used, it is still not suitable for unmanned vehicles’ localization, which need work in large-scale and long-term with high-precision required, so we introduce some Lidar localization systems with the pre-build map. Finally, we introduce some sensor fusion schemes for improving the localization stability. According to this structure, we have made some modifications to section 2.

Point 5: [98] "ICP" is an unidentified acronym.

Response 5: ICP is the acronym of Iterative Closest Point, it is a classical point cloud precise registration method. We have added the full name of ICP and the relevant reference. The detail can be seen in line 99 and line 612~613.

Point 6: [118] "fusion" should probably be "combining"

Response 6: Fusion is a more universal representation for multi-sensor data, the same usage can be found in reference [41] “Robust and Precise Vehicle Localization based on Multi-sensor Fusion in Diverse City Scenes”.

Point 7: [Algorithm 1: line 5: "F" is undefined.

Response 7: “F” is the error-state Kalman filter, and it has been defined in the first half of Algorithm 1, line 5.

Point 8: [176] what do you mean by "leaf size"?

Response 8: Leaf size is the size of each tile edge. This is a common expression, and it is also used in front line 179.

Point 9: [Section 3.3.1.] needs a figure/illustration to help present the discussion topic.

Response 9: Thanks for your suggestion. Section 3.3.1 introduces the algorithm of ground segmentation. It can use the same graph as section 3.3.2 to represent the geometric features when laser scans to the ground. We have added explanations to the line 221~223.

Point 10: [197] please identify the type of Velodyne sensor, there is more than one available on the market.

Response 10: Thanks for your suggestion. Velodyne VLP-32C is used in our experiment. We have added it in line 207. In addition, we also add the hardware system in section 4.1, line 385~393.

Point 11: [212] please rewrite the equation for clarity.

Response 11: We have changed the fractional representation from left-right to up-down, it can be seen in line 225.

Point 12: [221-222] this sentence is confusing to read.

Response 12: We are sorry for the missing explanation that the features can provide constraints for pose solving. We have change it to “We solve pose by the constraints provided by feature points, while the stable edge and surface features are less in the textureless scenes such as highway, so the constraints will be too less to increase the possibility of matching failure in such scenes.”. It can be seen in 234~236.

Point 13: [234] this new line should probably be part of the previous paragraph.

Response 13: Thanks for your suggestion. We are sorry for the mistake, and we have modified it.

Point 14: [247] what do you mean by "paperback"?

Response 14: We are so sorry for the wrong English expression. We mean the LiDAR installed horizontally. It has been changed to “horizontally installed” in line 262~263.

Point 15: [256] please discuss why the threshold is set to 0.03m.

Response 15: This is an empirical value. Judging from a large number of experimental results, we can get the best segmentation result when it is set as 0.03m, and it is proportional to the horizontal resolution of LiDARwhich is 0.02°for Velodyne VLP-32C. We have added the explanation into the manuscript in line 271~273.

Point 16: [Section 3.3.4.] please add an illustration/figure to help present the topic.

Response 16: Thanks for your suggestion. Section 3.3.4 introduces the proposed edge feature extraction algorithm, and we have added Figure 5, a technical flow chart to help present the topic, it can be seen in line 291.

Point 17: [268] "XOY" is undefined.

Response 17: XOY is the plane composed of x-axis and y-axis of the Lidar coordinate system. We have add it to the manuscript in line 285.

Point 18: [270] "RANSAC" is undefined.

Response 18: RANSAC is the acronym of random sample consensus, it is a commonly used iterative method for estimating the parameters of a mathematical model from a set of observed data containing outlies. We have added the full name into manuscript in line 287.

Point 19: [275] what do you mean by "lego-motion"?

Response 19: There are no “lego-motion”, do you mean “ego-motion”?  “ego-motion” is commonly used to represent the motion of the sensor between two instants of acquisition two adjacent frames data.

Point 20: [278] the sentence ends with "next", it seems to not be finished properly, please rewrite.

Response 20: We are sorry for the improper expression. We have changed it to “in the following paragraphs”. It can be seen in line 299.

Point 21: [315] Table 1 might do well with an extra column describing a computation metric because that is part of your discussion in the paper.

Response 21: Sorry, we cannot clearly understand the mean of “computation metric”. Table 1 shows the match method, the needed nearest points numbers, the calculation method of direction or normal vector for different feature types in frame-frame and frame-map matching. In addition, we have explained in line 333~335, that the computation complex of the three methods for vector calculation, PCA, direct assignment and formula (5)(6) are decreasing in turn. This is the reason why our proposed priori information considered matching method improves the matching efficiency and accuracy.

Point 22: [316] what does "L-M" mean?

Response 22: L-M is the acronym of Levenberg-Marquardt optimization method. It is a commonly used method for solving medium sized nonlinear optimization problems. We have added the full name of L-M and the relevant reference. The detail can be seen in line 63 and line 573.

Point 23: [338] Figure 5 can be redone to be more clear and straight forward in presenting the material. It's a little hard to absorb as is.

Response 23: Thanks for your suggestion. Figure 5 shows the pipeline of the proposed multi-group-step L-M optimization method, which maybe the too many symbols make it hard to absorb, we have deleted the input and output symbols of each step in the chart to ensure the clarity, and added explanation in line 350~360.

Point 24: [461] you identify that this work is "designed for high-level autonomous driving applications" but I didn't get that emphasis in the rest of the paper.

Response 24: In section 1 at the beginning of the article, we introduced the unmanned vehicles’ requirements for high-precision localization, as well as the current used localization schemes and the existing problems. The higher the level of autonomous driving, the higher the requirement for localization accuracy and stability. Here, we proposed a complete LiDAR-based localization system, and introduce a series of improvement, including map management method, segmentation-based feature extraction method, priori information considered category matching method, multi-group-step L-M optimization method, these improvements ensure the localization system can work stably and accurately. The proposed localization system provides essential localization service. Actually, the proposed localization system has been deployed in several unmanned vehicles for daily testing.

Point 25: [469] is "200Hz" the state of the art?  I didn't feel that it was based on the background in the paper.

Response 25: We are sorry that the inappropriate expressions have led to you misunderstand. The state-of-the-art means the localization accuracy, the keynote of this manuscript is to explain the proposed high-precision LiDAR-based localization system. We have used “the accuracy” instead of “which” to avoid confusion, it can be seen in line 526.

Point 26: [470] this half of the sentence is unclear and hard to read.

Response 26: We are sorry for the poor English expression. We have changed it to “achieve 3~125px in position and 0.05~0.1° in orientation RMS accuracy”, it can be seen in line 525~526.

Point 27: Additionally, it looks like the author put a lot of work into this effort but the paper didn't convey the sense of what was innovative and important about the work.

Response 27: This manuscript introduces a complete Lidar-based localization system, which provides essential high-precision and stable localization service for unmanned vehicles. The main contributions of the manuscript have been explained in line 59~71. After introducing the overall structure (section 3.1) of the system, the manuscript focus on the explanation of the improved algorithm (section 3.2, 3.3, 3.4, 3.5).

1. A novel efficient low-level semantic segmentation-based feature extraction algorithm is designed to extract multiple types of stable features from the online frames, including ground, road-curb, edge and surface. They ensure accurate pose estimation for frame-frame and frame-map matching.

2. A priori information considered category matching method and a multi-group-step L-M (Levenberg Marquardt) optimization algorithm are proposed, which can avoid most of the mismatching to improve the accuracy, and increase the efficiency by reducing the Jacobian matrix dimension.

3. An efficient priori map management is presented, the map is stored in tiles with overlap, and the local map surrounding the vehicle is dynamically loaded to save the computation resource.

4. A complex and complete vehicle localization system has been accomplished by integrating all modules reasonably, which can provide high-accuracy and real-time localization service for high-level unmanned vehicles in various challenging scenes and harsh weather.

By this series of improvements, this system can provide real-time and stable localization results, and the accuracy outperform the state-of-the-art. It not only can solve the problems existing in the currently used scheme, but also greatly reduces the cost and further accelerates the large-scale landing of the unmanned vehicles.

Reviewer 4 Report

Some notations are not explained at its first appearance or not consequent, e.g. line 215. The authors state that they use difference treshold for ground segmentation, because other methods were slow to adapt their system. However, some kind of preprocessing must be needed to find ground seed points. How is it done? The authors states multiple times that the error of their system less than previous state of the art systems’, but there is no report about the performance of these previous systems. More importantly, to compare the performances, other systems need to be evaluated on the author’s dataset (or the authors’ method on the datasets of the others). This comparison part has to be correctly done. The authors state their system can operate in harsh weather environments, they mention e.g. fog in the conclusion. However, no evidence is provided to support their statement (in the evaluation part only rainy data and its results are presented) or explanation why their method should work in these circumstances.

Author Response

Thanks for your hard work on our manuscript. Your constructive comments and suggestions are very valuable for improving the manuscript. The revised manuscript has been uploaded. Here is the point-by-point response for your comments.

Point 1: Some notations are not explained at its first appearance or not consequent, e.g. line 215.

Response 1: Thanks for your suggestion. First, we are so sorry for writing  as , which represents the difference in  direction between two adjacent points in the th column. Second, the meanings of ,  are consistent with , so we have modified them with the same notations to avoid confusion, the detail can be seen in line 225~228. And we have checked all the manuscript for other un-explained notations.

Point 2: The authors state that they use difference threshold for ground segmentation, because other methods were slow to adapt their system. However, some kind of preprocessing must be needed to find ground seed points. How is it done?

Response2: Thanks for your suggestion. We are sorry for the unclearly explain. In the reference [47], “The Ground Segmentation of 3D LIDAR Point Cloud with the Optimized Region Merging”, they calculate the normal difference and gradient, then extract the ground by region growing, which need seed points before beginning growing. While our proposed method takes advantage of the geometric characteristics of the ground points and extracts ground points by a vertical angle, without the need for seed points. For a horizontally installed LiDAR, when the laser scanning on the flat ground surface, the difference in  direction is far less than  directions between two adjacent points in the same column. Consider such a geometric property, we define a term to represent the vertical angle between two adjacent points in the same column. Our proposed method can extract ground point cloud efficiently, and the computation complexity is very low. We have added the explain into the manuscript in line 221~224.

Point 3: The authors state multiple times that the error of their system less than previous state of the art systems’, but there is no report about the performance of these previous systems. More importantly, to compare the performances, other systems need to be evaluated on the author’s dataset (or the authors’ method on the datasets of the others). This comparison part has to be correctly done.

Response 3: Thanks for your suggestion. The reference [41], “Robust and Precise Vehicle Localization based on Multi-sensor Fusion in Diverse City Scenes”, is the paper which we called previous state-of-the-art. In the existing reports, it is the most complete and accurate Lidar-based localization system, where the  can achieve 5~250px RMS accuracy, and  is about 250px RMS. We have added a description in line 501~504. The always used method to evaluate the performance of two algorithms is using the same dataset, but there are very few complete reports about high-precision LiDAR-based localization system for automated urban driving, most of the research on LiDAR mapping and localization is a small-scale low-precision SLAM system that is not based on pre-build map. Reference [41] is a relatively advanced and complete LiDAR-based localization system for unmanned vehicles, they use the intensity information for frame-frame and frame-map LiDAR matching, and integrate IMU and GNSS data, but they don’t release the dataset. Because the intensity information of LiDAR data is used, which requires a strict and complex reflection intensity calibration for each LiDAR device, so our data is not applicable for their method. While we all use high-precision INS and a series of post-processing to obtain very high precision as ground truth to evaluate the precision of our localization system, that is to say, the reported errors are absolute values rather than relative values, so we can compare our accuracy with them to evaluate the quality of the localization system. What’s more, we are also trying to make and release our LiDAR localization evaluation dataset to provide more convenience for future researchers and accelerate technological progress.

Point 4: The authors state their system can operate in harsh weather environments, they mention e.g. fog in the conclusion. However, no evidence is provided to support their statement (in the evaluation part only rainy data and its results are presented) or explanation why their method should work in these circumstances.

Response 4: Thanks for your suggestion. The harsh weathers, such as rain and fog, have similar effects on LiDAR data, which reduce the LiDAR measurement range and bring some noisy points, and this phenomenon is more serious in rainy days. While our proposed segmentation-based feature points extraction algorithm can effectively filter out the influence of noise and make full use of the limited point cloud information to stably solve out high-precision pose. Therefore, a section data of rainy day with worse quality is selected for comparison. We have added the explanations into the manuscript in line 469~475 to avoid confusing the readers.

Round 2

Reviewer 3 Report

Firstly, I'd like to say that the revised version is much easier to read and I feel I'm more easily able to follow the authors intent.  Additionally, the figures are much easier to read and the paper's meaning is more clear to the reader  

There are some very minor things:

XOY is not defined in the paper.  The authors explained it in the response to prior comments but should be included for ease of readability.. 

There are some minor english grammer errors such as in line [415] "it" should probably be "is", and in line [456] "thecluttered" should be two words and "leafs" should be "leaves".  I would reccomend another english edit/proofread before publication.  

Author Response

Firstly, I'd like to say that the revised version is much easier to read and I feel I'm more easily able to follow the authors intent. Additionally, the figures are much easier to read and the paper's meaning is clearer to the reader  

Thanks for your hard work and affirmation on our manuscript. Your constructive comments and suggestions have help us to improve the manuscript’s quality. The revised manuscript has been uploaded. Here is the point-by-point response for your comments.

Point 1: XOY is not defined in the paper. The authors explained it in the response to prior comments but should be included for ease of readability. 

Response 1: Thanks for your suggestion. We have added the detailed definition of XOY plane into the manuscript, which can be seen in line 289~290.

Point 2: There are some minor English grammar errors such as in line [415] "it" should probably be "is", and in line [456] "thecluttered" should be two words and "leafs" should be "leaves". I would recommend another English edit/proofread before publication. 

Response 2: Thanks for your careful review. We are so sorry for these English grammar errors. We have used a professional English editing service from MDPI in last round of revision, and perform English proofread one more time to check out the grammar and spelling mistakes.

Reviewer 4 Report

I accept all of the answers of the authors except one. The ground segmentation method used by the authors still needs some more clarification. They state, there are no seed points in their ground segmentation, so all the points are classified to be ground or not based on thresholding a geometric feature (which is basically elevation difference of consecutive points). In this case, any other flat surface’s inner points (which fulfill the elevation difference criterion) would be classified as ground.

Author Response

Point 1: I accept all of the answers of the authors except one. The ground segmentation method used by the authors still needs some more clarification. They state, there are no seed points in their ground segmentation, so all the points are classified to be ground or not based on thresholding a geometric feature (which is basically elevation difference of consecutive points). In this case, any other flat surface’s inner points (which fulfill the elevation difference criterion) would be classified as ground.

Response 1: Thanks for your suggestion. Our proposed ground segmentation algorithm is based on a geometric feature. When Lidar scans on the ground, the difference in the z direction is far less than the x and y directions between two adjacent points in the same column, and the elevation difference is close to 0. So, we define and calculate the vertical angle of the two points, then use the vertical angle and the difference in z direction as the threshold (line 229~230) to extract the ground points. It is vertical angle, not elevation difference, that plays the key role in ground segmentation.

As you said, our algorithm extracts not only road surface but also flat ground, it can be seen in Figure7 (b), it is also explained in line 429~430. These flat ground points can paly the same role as road surface points in optimizing pose, and they are also needed in our localization system. So, we call the algorithm ground segmentation rather than road segmentation. In order to avoid misunderstanding, we have added some explanation in line 224~225, 230~231, 234~236.
